# Uniaxial Strain Dependence on Angle-Resolved Optical Second Harmonic Generation from a Few Layers of Indium Selenide

**DOI:** 10.3390/nano13040750

**Published:** 2023-02-16

**Authors:** Zi-Yi Li, Hao-Yu Cheng, Sheng-Hsun Kung, Hsuan-Chun Yao, Christy Roshini Paul Inbaraj, Raman Sankar, Min-Nan Ou, Yang-Fang Chen, Chi-Cheng Lee, Kung-Hsuan Lin

**Affiliations:** 1Institute of Physics, Academia Sinica, Taipei 115201, Taiwan; 2Department of Physics, National Taiwan University, Taipei 10617, Taiwan; 3Department of Physics, Tamkang University, Tamsui, New Taipei 251301, Taiwan

**Keywords:** second harmonic generation, strain, InSe, first–principles calculation

## Abstract

Indium selenide (InSe) is an emerging van der Waals material, which exhibits the potential to serve in excellent electronic and optoelectronic devices. One of the advantages of layered materials is their application to flexible devices. How strain alters the electronic and optical properties is, thus, an important issue. In this work, we experimentally measured the strain dependence on the angle-resolved second harmonic generation (SHG) pattern of a few layers of InSe. We used the exfoliation method to fabricate InSe flakes and measured the SHG images of the flakes with different azimuthal angles. We found the SHG intensity of InSe decreased, while the compressive strain increased. Through first–principles electronic structure calculations, we investigated the strain dependence on SHG susceptibilities and the corresponding angle-resolved SHG pattern. The experimental data could be fitted well by the calculated results using only a fitting parameter. The demonstrated method based on first–principles in this work can be used to quantitatively model the strain-induced angle-resolved SHG patterns in 2D materials. Our obtained results are very useful for the exploration of the physical properties of flexible devices based on 2D materials.

## 1. Introduction

Two-dimensional (2D) materials have drawn much attention since the discovery of graphene in 2004 [1]. Subsequently, graphene and the subsequent derivative 2D materials have been demonstrated for many applications such as in electronic and optoelectronic devices [2,3]. Since van der Waals materials can be mechanically exfoliated and transferred to arbitrary substrates, including bendable polymeric or plastic substrates, studies of the strain-dependent properties of 2D materials are therefore beneficial for the development of flexible devices [4,5]. Indium selenide (InSe) is an emerging star in van der Waals semiconductors [6] due to its superior properties such as ultrahigh mobility [7,8] and large elastic deformability [9,10]. It has been applied to bendable photodetectors with a good performance and a broad spectral response [11]. For the application of flexible devices, it is important to understand how strain affects the relevant physical properties. For example, InSe was found to exhibit the ultrasensitive tunability of the direct bandgap by applying uniaxial strain [12,13]. The variation in optical properties under strain could be utilized as a tool to map the intensity distribution of strain such as photoluminescence spectroscopy and micro-Raman spectroscopy [14,15]. Recently, optical second harmonic generation (SHG) techniques have been demonstrated to be a powerful and unique tool for mapping strain vectors. In addition to the intensity distribution of strain, the direction of the strain distribution can also be mapped by analyzing the angle-resolved SHG patterns [16].

SHG is a second-order nonlinear process that only occurs in crystals without inversion symmetry. In a thick crystal, the conversion efficiency of SHG is maximized in the condition of phase matching. For thin crystals with a thickness below 100 nm, a few methods have been proposed to increase the efficiency of SHG recently [17,18,19]. For example, an optical interference layer at the bottom of the thin crystals enhances the SHG when the thickness of the interference layer is designed in the phase condition for the constructive interference of both the fundamental and second harmonic frequencies [17]. Nanostructured crystals serving as dielectric nanoantennas for second harmonic emissions have also been demonstrated [18,19]. While the Mie resonant modes are matched with the fundamental or second harmonic frequencies, the efficiency of SHG is increased. The nonlinear optical process highly depends on the motion of the electrons in the lattice structure. It is thus capable of determining the orientations of single crystals [20]. For example, monolayer transition metal dichalcogenide (TMD) materials, such as ${\mathrm{MoS}}_{2}$, ${\mathrm{WS}}_{2}$, and ${\mathrm{WSe}}_{2}$ possess hexagonal structures with a point group of $D_{3h}$. When a linearly polarized laser is normally incident to a TMD monolayer, SHG light with the same polarization can be recorded as a function of the rotation angles of the TMD. This angle-resolved SHG pattern exhibits six-fold rotational symmetry, and the maximum SHG intensity occurs during laser polarization along the arm-chair (AC) direction [21,22,23]. If strain is applied to the hexagonal TMD monolayers, the crystal symmetry reduces, and the angle-resolved SHG patterns vary accordingly [16,24,25]. Since not only the intensity of SHG, but also the symmetry of the angle-resolved SHG patterns are analyzed, it is possible to monitor the local strain vector. InSe possesses high mobility, which is an advantage over the TMDs in certain device applications. It possesses a periodic honeycomb lattice in three specific phases (β, γ, and ε) based on different stacking sequences [26]. SHG is allowed in monolayers and a few layers of InSe in γ phase/ε phase due to their non-centrosymmetric structures [27,28,29,30].

In this work, we investigated the strain dependence on angle-resolved SHG from mechanically exfoliated γ-InSe flakes. We mapped the SHG images of the strained InSe flakes with different rotation angles and studied the angle-resolved SHG pattern. Overall, the SHG intensity decreased with increasing compressive strain. By performing first–principles electronic structure calculations within the density functional theory, we also studied the uniaxial strain dependence on SHG susceptibilities. The calculated susceptibility decreases with increasing compressive strain, which agrees with the experimental results. We demonstrate a method based on first–principles calculations to quantitatively model the strain-induced angle-resolved SHG patterns in InSe flakes, which is applicable to other 2D materials. This model is useful to analyze angle-resolved SHG patterns for the strain mapping of 2D materials.

## 2. Methods

### 2.1. Growth of InSe Single Crystals and Their Characterizations

InSe crystals were grown by the vertical Bridgman method. We purchased 99.999% pure Indium (In) and Selenide (Se) chemicals from Sigma Aldrich. The growth of the InSe single crystals was performed in quartz ampoules by mixing In and Se powders under a vacuum of 10^−4^ Pa. The ampoules were placed in a horizontal furnace at 550 °C for two days to obtain a partially mixed crystal. Later, the mixed crystals were used to grow the pure InSe single crystal with the help of a vertical Bridgman setup. The furnace during the Bridgman process was maintained at a high temperature of about 850 °C. The mixed crystal in the ampoule was hung and heat treated at 850 °C for 24 h; once the mixed crystal melted entirely in the ampoule, the ampoules were lowered through a temperature gradient of 1 °C at a rate of 0.1 mm/h to obtain a large-sized single crystal. Using this procedure, we obtained over 5 cm long and 1 cm in diameter InSe single crystals in bulk. 

Powder X-ray diffraction (PXRD) was employed to study the lattice phases. The result of the Rietveld refinement of X-ray diffraction was carried out by the software GSAS, while the high orientation of the powdered InSe was taken into account for the refinement. The lattice structure was determined as γ-InSe (R3m) with lattice constants a = b = 4.006 Å, and c = 24.969 Å, respectively. The crystal phase was also confirmed by oblique incident SHG measurements [31]. Although SHG is allowed in both the γ phase and ε phase due to their non-centrosymmetric structures, their different stacking orders lead to distinguishable z-related SHG susceptibilities, which are measurable through the angle-resolved p-polarized SHG (p-out) with an oblique incidence of the p-polarized fundamental light (p-in). The p-out p-in SHG angle-resolved pattern of our InSe crystal manifested 3-fold rotational symmetry, indicating the feature of γ phase because the p-out p-in angle-resolved SHG pattern of ε phase InSe exhibits 6-fold rotational symmetry.

The sample structure of γ-InSe is shown in Appendix A. The cross-section view along the [010] direction in Appendix A indicates how the monolayers of InSe are stacked in an ABC manner. The top view of γ-InSe along the [001] direction is shown in Appendix A, where the AC and zigzag (ZZ) directions are labelled with black arrows. The bulk γ-InSe crystal belongs to the $C_{3v}$ point group and exhibits a non-centrosymmetric structure.

### 2.2. Sample Preparation and Strain Engineering

Flakes of InSe in a few layers were prepared by the mechanical exfoliation of the γ-InSe crystal with polydimethylsiloxane (PDMS). Multiple flakes were subsequently transferred to a polyethylene naphthalate (PEN) flexible substrate, with an SU-8 photoresist film on top. The thickness of the PEN substrate was about 125 μm, and the thickness of SU-8 photoresist film was in the order of 1 μm. On top of the PEN substrate with the SU-8 photoresist film, the InSe flakes were examined using an atomic force microscope (AFM). Appendix A shows the AFM image around the edge of an InSe flake. Four line profiles of height were analyzed, as shown in Appendix A, which reveal that the thickness is in the range between 7.3 and 8.7 nm. The thickness of the InSe flake we studied in this work is ~8 nm, which corresponds to ~10 layers.

We referred to the experimental configuration in the literature [32] to design a device for applying uniaxial strain. As shown in Appendix A, the strain device was combined with two sliding stages and screws. The flexible PEN substrate was placed between the two sliding stages. Uniaxial strain was applied by curving the PEN substrate via spinning the screws to move the two sliding stages toward each other. Appendix A shows the schematic of the curved substrate under uniaxial strain. The uniaxial strain ε in this work was determined following the relationship ε=d/2R, where d is the thickness of the substrate and R is the radius of curvature of the substrate. In the experiments, 125 μm was used for d. To obtain the radius of curvature, the cross-section images of the curved PEN substrate on the strain device were recorded using a digital camera. The error of fitting the radius of curvature and retrieving the strain was within 3%.

### 2.3. SHG Measurements

The schematic of the experimental setup is shown in Appendix A. The ultrafast pulses centered at ~790 nm were generated by a Ti:sapphire laser at 80 MHz repetition rate (Newport, Tsunami). The bandwidth of the pulses was ~12 nm. A set of a half-wave plate (HWP) and a polarizing beam splitter (PBS) cube was used to control the power to prevent damage occurring to the samples. A pulse compressor was used to compensate the dispersion of the transparent optical components to maximize the efficiency of SHG from the samples. The beam was directed through a galvo-mirror system and a long working distance objective lens (20×, NA = 0.35) to scan the samples in two dimensions. A linear polarizer (LP) was used to purify the polarization of the incident beam. A dichroic mirror was used to separate the fundamental incident beam and SHG beam from the sample. The SHG photons were sensed by a single-photon-counting photomultiplier tube (PMT). A 400 nm bandpass filter (BPF) was placed in front of the PMT to eliminate the light at the fundamental frequency. A linear polarizer was placed in front of the bandpass filter, which was rotated to selectively measure the SHG light with polarization that was parallel or perpendicular to the polarization of the fundamental incident light. A motorized rotational stage (Newport, URS150BCC) was on an inverted microscope for recording the angle-resolved SHG images with a stepping angle of 10 degrees.

### 2.4. Theoretical Calculations

To theoretically obtain the geometrical structures of InSe under strain and calculate the SHG susceptibilities of the strained InSe, we performed first–principles calculations based on the density functional theory (DFT) using the OpenMX code [33], where generalized gradient approximations (GGA) [34], norm-conserving pseudopotentials [35], and optimized pseudoatomic basis functions [36] were adopted. We constructed a lattice of the γ phase InSe, with a unit cell consisting of three InSe monolayers in an ABC stacking order, as shown in Appendix A. The x-axis was set to be the AC direction. For the atomic basis functions, three, two, and two optimized radial functions were allocated to the *s*, *p*, and *d* orbitals, respectively. We adopted a cutoff radius of 7 bohr for the In atom, denoted as In7.0-s3p2d2. Similarly, for the Se atom, Se7.0-s3p2d2 was adopted. A cutoff energy of 500 Ry was used for numerical integrations and for the solution of the Poisson equation. An 8 × 8 × 2 *k*-point sampling was used. The structures were fully relaxed without considering spin-orbit coupling at the specified strain, that is, only the components of the lattice vectors associated with the desired strain were fixed during the calculations. For the unstrained ground-state structure, no constraint was applied to the lattice vectors. Once the optimized lattice constants were obtained, the atomic positions were relaxed again with the effect of spin-orbit coupling. All of the atomic forces in the resulting lattice satisfied the criterion of being smaller than 0.0001 Ha/bohr. The electronic self-consistent field (SCF) criterion was set to 10^−9^ Ha, which guaranteed that the difference between the band energies in the last two consecutive SCF steps was smaller than the criterion.

The spectra of SHG susceptibilities were calculated via the momentum matrix elements using the basis of the adopted pseudoatomic basis functions [37], following the formula derived by Sipe et al. for the independent-particle approximation [38,39]. A 60 × 60 × 8 k-point sampling was used for obtaining the SHG susceptibilities with a broadening parameter of η=0.1 eV. The electronic temperature was set to 300 K for all of the SCF and SHG calculations based on the Fermi–Dirac distribution function. Due to the band gap error in the DFT calculations [40], the calculated band gaps were underestimated, and a scissor operator with 0.6487 eV was applied to increase the band gaps by shifting the eigenvalues of unoccupied Kohn–Sham orbitals. The modification to the momentum matrix elements, which was accompanied by the energy shift, was also taken into account in the SHG calculations [39]. 

## 3. Results and Discussion

Figure 1 reveals the scanning optical images of unstrained InSe flakes. To confirm that the PMT signals in Figure 1 were SHG signals from the InSe flakes, we measured the spectrum of the collected light in front of the PMT, as shown in Appendix A. The Gaussian-like feature centered at 394 nm confirmed the SHG of the incident light pulse centered at ~790 nm. We also conducted power-dependent measurements to confirm the SHG process, as shown in Appendix A. The fitting line with a slope of ~2 in the log–log scale confirmed the quadratic dependence of SHG intensity on the fundamental laser power. For the SHG images in Figure 1, the polarizers were set so that the polarization of the detected SHG was parallel to the polarization of the incident light. While the samples were rotated from −20 to 30 degrees, the optical intensity of the biggest flakes among the three flakes increased to a maximum at 0 degree, shown in Figure 1c, and decreased to a minimum at 30 degrees, shown in Figure 1f. To further quantitatively analyze the angle-resolved SHG intensity, we summed up the signals over the biggest flakes in the images of different rotational angles. The angular dependence on the SHG intensity of parallel polarization (denoted as $I_{∥}^{\left(2\omega \right)}$) from the biggest InSe flake is revealed by the blue squares in Figure 2. Because the hexagonal γ-InSe exhibits the point group of $C_{3v}$, the angle-resolved $I_{∥}^{\left(2\omega \right)}$ under normal incidence also reveals six-fold rotational symmetry. The maximum intensity occurs at the angle of the incident light polarization along the AC direction [41]. The angle-resolved $I_{∥}^{\left(2\omega \right)}$ was fitted with $\cos^{2}3\theta$, as shown by the blue line in Figure 2. Here, θ is defined as the angle of AC direction with respect to the linear polarization of the incident laser beam. Angle-resolved analysis is capable of determining the crystal orientations of different flakes on the substrate. For example, while the SHG intensity of the upper left flake in Figure 1a was traced for different rotational angles in Figure 1a–f, the maximum SHG intensity occurs at 10 degrees in Figure 1d. In contrast to the maximum SHG intensity of the biggest flake occurring at 0 degree in Figure 1c, the orientation of the upper left flake was rotated by 10 degrees with respect to that of the biggest flake. We also conducted mapping measurements with SHG polarization perpendicular to the polarization of the incident light (denoted as $I_{⊥}^{\left(2\omega \right)}$). For a similar angle-resolved analysis of the biggest flake, the orange points in Figure 2 depict the experimental data in the polar plot of $I_{⊥}^{\left(2\omega \right)}$, which manifest a thirty-degree-rotated pattern of $I_{∥}^{\left(2\omega \right)}$, and they are fitted with $\sin^{2}3\theta$.

We further applied compressive uniaxial strain on the samples along the AC direction (θ=0°) of the biggest InSe flakes and mapped the images of $I_{∥}^{\left(2\omega \right)}$ as a function of rotational angles. The polar plots of $I_{∥}^{\left(2\omega \right)}$ under different strain levels (−0.18%, −0.36%, −0.53%, and −0.83%) are shown in Figure 3. The best fitting lines with $\cos^{2}3\theta$ are shown to guide the eyes. Overall, the intensity of SHG decreases with increasing compressive strain.

In order to obtain more insight into strain-induced angle-resolved SHG patterns, we theoretically calculated the uniaxial strain dependence on SHG susceptibilities. First, we used the first–principles method to calculate the band structures of InSe under different compressive strains. The strain was applied along the AC direction, which is defined as the x direction. Subsequently, the spectra of SHG susceptibilities were calculated following the method described in Section 2.4 and are shown in Appendix A. The SHG susceptibilities are dependent on the photon energy. Here, we focus on the non-zero SHG susceptibilities at 1.55 eV, which is the photon energy of the incident light in the experiments. Figure 4a shows the strain dependence on $\left|\chi^{\left(2\right)}\right|$ at 1.55 eV. The real parts, imaginary parts, and phases of non-zero SHG susceptibilities at 1.55 eV are shown in Appendix A. When the compressive strain increases from 0% to −2%, all of the non-zero components of $\left|\chi^{\left(2\right)}\right|$ decrease. These trends agree with the experimental results that the SHG intensity decreases with increasing strain. We developed a model based on these first–principles results to quantify the rotational angle dependence on the polarized SHG intensity and to compare them with the experimental data. The second-order polarization $P^{\left(2\omega \right)}$ can be expressed as
(1)$$P_{i}^{\left(2\omega \right)}\propto \sum_{jk}\chi_{ijk}^{\left(2\right)}E_{j}E_{k}.$$
$\chi_{ijk}^{\left(2\right)}$ is the second-order susceptibility tensor component and $E_{j}$ ($E_{k}$) is the electric field. The indices i, j, and k can be 1, 2, and 3 corresponding to x, y, and z in Cartesian coordinates, respectively. Because $E_{j}$ and $E_{k}$ have the same frequencies *ω* in SHG, the second-order susceptibility possesses intrinsic permutation symmetry, which is $\chi_{ijk}^{\left(2\right)}=\chi_{ikj}^{\left(2\right)}$. By considering this permutation rule, the third-rank tensor $\chi^{\left(2\right)}$ for SHG is contracted to a 3×6 matrix with element $d_{im}$. The index m can be 1 = xx, 2 = yy, 3 = zz, 4 = yz, zy, 5 = xz, zx, and 6 = xy, yx. Since the unstrained γ phase InSe belongs to the $C_{3v}$ point group, its d-matrix has eight non-zero components, as follows [20].
(2)$$d=\left[\begin{array}{cccccc} d_{11} & d_{12} & 0 & 0 & d_{15} & 0\\ 0 & 0 & 0 & d_{24} & 0 & d_{26}\\ d_{31} & d_{32} & d_{33} & 0 & 0 & 0 \end{array}\right],$$
where $-d_{11}=d_{12}=d_{26}$, $d_{15}=d_{24}$, and $d_{31}=d_{32}$. Because the fundamental beam is normally incident to InSe, we consider the polarization of light on the x–y plane with θ respect to the x-axis. The electric field vector can be written as [20]
(3)$$E=\left[\begin{array}{c} \cos^{2}\theta \\ \sin^{2}\theta \\ 0\\ 0\\ 0\\ 2\sin\theta \cos\theta \end{array}\right].$$

The parallel SHG intensity for unstrained InSe is
(4)$$I_{∥}^{\left(2\omega \right)}\propto {\left|P_{x}\cos\theta +P_{y}\sin\theta \right|}^{2}={\left|d\cos3\theta \right|}^{2},$$
where $d=d_{11}=-d_{12}=-d_{26}$. Because both the SHG and fundamental polarizations are on the x–y plane, the z-related components, namely $d_{15},\;d_{24},\;d_{31},\;d_{32},$ and $d_{33}$, do not contribute to the signals. Due to the relationship between the other three non-zero components, $-d_{11}=d_{12}=d_{26}$, $I_{∥}^{\left(2\omega \right)}$ is thus only determined by one parameter d in Equation (4). 

Since the variation in SHG susceptibilities is linear to the compressive strain, the strain effect on the SHG susceptibilities could thus be modelled as follows.
(5)$$d_{11}^{strain}=d_{11}^{unstrain}+A\varepsilon ,\;d_{12}^{strain}=d_{12}^{unstrain}+B\varepsilon ,\;d_{26}^{strain}=d_{26}^{unstrain}+C\varepsilon .$$

The coefficients A,B, and C are complex numbers and specify the linear deviations of complex SHG susceptibilities $d_{11},\;d_{12},$ and $d_{26}$ with respect to strain, respectively. Since $d_{11}^{unstrain}=-d_{12}^{unstrain}=-d_{26}^{unstrain}=d$, the angle-dependent SHG intensity under uniaxial compressive strain is derived as
(6)$$I_{∥}^{\left(2\omega \right)}\propto {\left|d\cos3\theta +\varepsilon \left[A\cos^{3}\theta +\left(B+2C\right)\sin^{2}\theta \cos\theta \right]\right|}^{2}.$$
d is the calculated SHG susceptibility without strain. The coefficients A, B, and C were obtained by generating linear fits of the calculated strain dependence on the SHG susceptibilities shown in Appendix A. The blue line and the red line in Figure 4b reveal the calculated angle-resolved $I_{∥}^{\left(2\omega \right)}$ for ε=0 and ε=−20% in Equation (6), respectively. Here, a large strain value of −20% was intentionally used to highlight the variation in the angle-resolved SHG pattern. Compared with the pattern without strain, the overall intensity reduces, but it exhibits anisotropy due to the strain. The rotational symmetry is reduced from six-fold to two-fold symmetry because the uniaxial compressive strain along the AC direction lowers the crystal symmetry of γ-InSe from the point group of $C_{3v}$ to the point group of $C_{2v}$. The two petals at 0° and 180°, which mean that the laser polarization is along the uniaxial strain direction, reduce more than the other four petals do. In Equation (6), the dcos3θ term, which is a function with six-fold rotational symmetry, describes the angle-resolved $I_{∥}^{\left(2\omega \right)}$ without strain, as shown by the blue line in Figure 4b. The modification due to the strain effect is found in the term $\left[A\cos^{3}\theta +\left(B+2C\right)\sin^{2}\theta \cos\theta \right]$, which is a function with two-fold rotational symmetry. Therefore, the modification due to the strain term reduces the symmetry of the angle-resolved $I_{∥}^{\left(2\omega \right)}$, as shown by the red line in Figure 4b.

To quantitatively understand the appropriateness of the aforementioned model, we used the strain values in the experiments to calculate the angle-resolved $I_{∥}^{\left(2\omega \right)}$ in Equation (6) for a comparison with the experimental data. Note that d, A, B, and C are derived from the results of first–principles calculations in Figure 4a and are not the fitting parameters. With only a scaling factor $f^{2}$, which is multiplied on the right-hand side of Equation (6), the blue line in Figure 5a reveals the best fit curve for ε=0. By using the same scaling value of $f^{2}$, the red line depicts $I_{∥}^{\left(2\omega \right)}$ for ε=−0.83%. Although the intensity of both the experimental and theoretical data drops due to the strain, that of the experimental data drops more, as shown in Figure 5a. To quantitatively evaluate the deviation of theoretical curves from the experimental results, the r-squared values of each theoretical curve for ε= −0.18%, −0.36%, −0.53%, and −0.83% were calculated as 0.992, 0.987, 0.985, and 0.968, respectively, as shown in Appendix A. The r-squared value decreases with an increasing strain value, indicating that the theoretical curves deviate more from experimental results under higher strain. In other words, the model based on the first–principles calculation underestimates the strain-induced variation in the SHG susceptibilities.

The model should be modified to appropriately fit the experimental data. Here, we additionally introduced a correction parameter $f_{2}$ and replaced ε in Equation (6) with $f_{2}\varepsilon$. By fitting the parameter $f_{2}$ = 4.48, we obtained the maximum summation of the r-squared values for all of the data with different strains. The r-squared values of each modified theoretical curve for ε= −0.18%, −0.36%, −0.53%, and −0.83% are 0.993, 0.996, 0.996, and 0.995, respectively, as shown in Appendix A. The fitting results for all of the experimental data are also depicted in Figure 5b. By only using a correction parameter, the results from the model based on first–principles calculations quantitatively agree well with the experimental data.

## 4. Conclusions

We mapped the SHG images of γ-InSe flakes under uniaxial strain. By recording the SHG intensity mapping of InSe flakes with different azimuthal angles, the angle-resolved SHG patterns were obtained. We also performed first–principles calculations to study the strain dependence on the SHG susceptibilities and the corresponding angle-resolved SHG patterns. Both the experimental and calculated results indicate that the SHG intensity decreased with increasing uniaxial strain on the InSe flake. Our experimental investigation and theoretical model provide a way to analyze angle-resolved SHG patterns for the strain mapping of γ-InSe flakes, which is very useful for better understanding the physical properties of 2D materials implemented in flexible devices. The modeling method proposed in this work is expected to be applicable to investigations of the strain-dependent SHG in other 2D materials.

## Figures and Tables

**Figure 1 nanomaterials-13-00750-f001:**
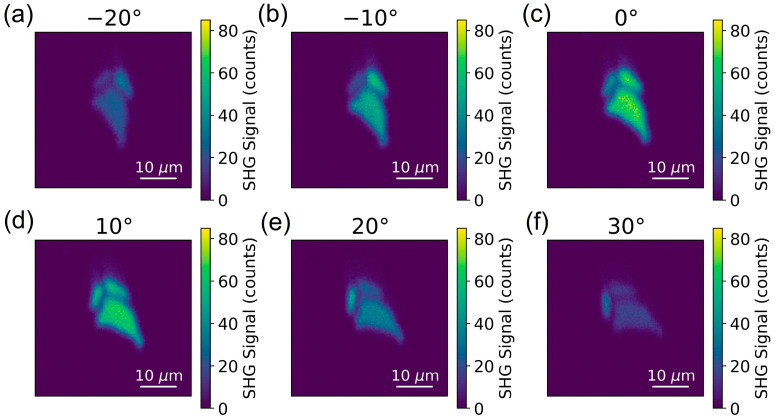
Images of SHG with polarization parallel to the incident light polarization for rotational angles (**a**–**f**) from −20° to 30°.

**Figure 2 nanomaterials-13-00750-f002:**
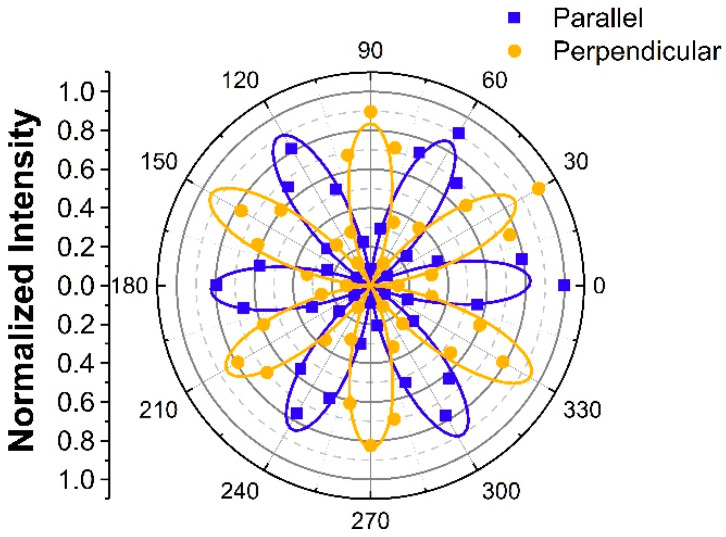
Normalized angle-resolved SHG patterns for SHG polarization parallel (blue squares) and perpendicular (orange dots) to the incident light polarization. The blue and orange solid lines are $\cos^{2}3\theta$ and $\sin^{2}3\theta$ curves, respectively.

**Figure 3 nanomaterials-13-00750-f003:**
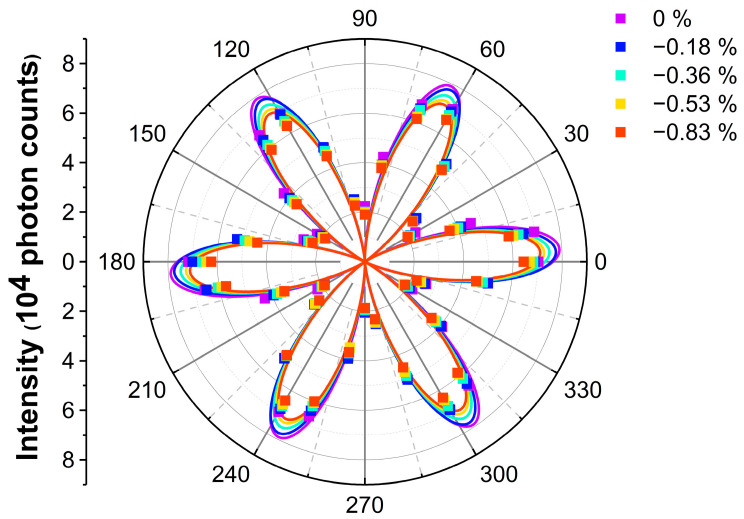
Angle-resolved SHG of InSe under different strain levels. The dots depict the experimental data and are fitted with $\cos^{2}3\theta$, which is shown in solid lines to make it easier to see.

**Figure 4 nanomaterials-13-00750-f004:**
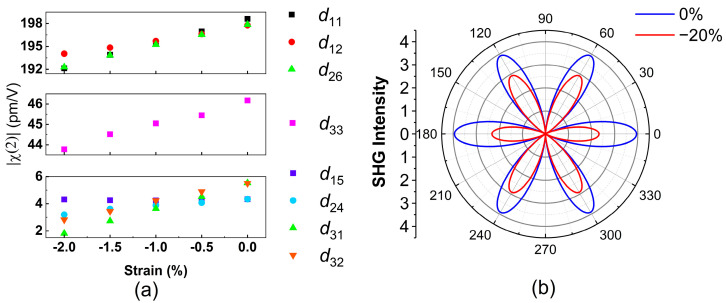
(**a**) Calculated nonzero $\left|\chi^{\left(2\right)}\right|$ components at photon energy 1.55 eV at different uniaxial strain levels. (**b**) Angle-resolved $I_{∥}^{\left(2\omega \right)}$ pattern under 0% and −20% strain derived from the calculated $\chi^{\left(2\right)}$ at 1.55 eV.

**Figure 5 nanomaterials-13-00750-f005:**
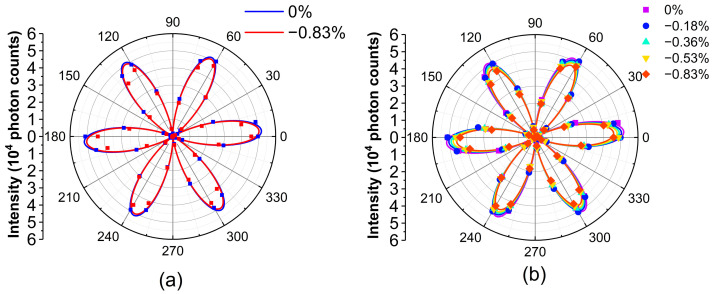
(**a**) The experimental results in dots and the calculated curves in solid lines from the model based on the first-principles methods without a correction parameter. (**b**) The calculated curves with a correction parameter to linearly modify the strain term.

## Data Availability

The data that support the findings of this study are available from the corresponding author upon reasonable request.

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
