# Peer review of "Uniaxial Strain Dependence on Angle-Resolved Optical Second Harmonic Generation from a Few Layers of Indium Selenide"

_nanomaterials, 2023, doi:10.3390/nano13040750_

Round 1

Reviewer 1 Report

The manuscript entitled “ Uniaxial Strain Dependence on Angle-resolved Optical Second Harmonic Generation from Indium Selenide Few-Layers” by  Zi-Yi Li et al. describes the results of the experimental and theoretical studies of second harmonic generation in exfoliated flakes of Indium Selenide. They demonstrate that the symmetry of the nonlinear optical response changes under the application of the uniaxial strain, similarly to observed previously for other materials.

While in general the presented results are interesting, still the paper have to be modified. Below is the list if inperfections of the manuscript that should be considered.

  • The consequence of presenting the results in the manuscript seems not optimal, as first the authors state that they use definite functions to fit the obtained anisotropic SHG intensity curves, and justify the approximation functions much later. So for the readers this is inconvenient.
  • The authors use the two notations for the second-order susceptibility, while it seems that using only one of them would make the understanding of the research easier.
  • The authors describe their results on SHG in terms of the nonlinear susceptibility terms – do they consider complex number or just real? And why? Please explain in the text. Do they analyze the possible variation of phase of the nonlinear susceptibilities with the strain increase and the relevant changes in the SHG response?
  • Please provide the definition of strain that is specific for the described studies.
  • What is the accuracy for the estimation of strain  -  in theory and experiment?

Reviewer 2 Report

It is a well performed theoretical and experimental study on SHG and its dependents on strain. 

there is good match between modelling and experiment. 

One issue could be clarified. 2D nanomaterials can be placed on different substrates. there is a possibility to enhance SHG by using correct phase conditiosn for the reflected light (

  • Optics Express
  • Vol. 30,
  • Issue 12, 
  • pp. 22161-22177 
  • (2022) https://doi.org/10.1364/OE.460118) 
  •  

How such effect would change SHG from 2D materials. related issue is placing of several 2D flakes at different orientation. 

Reviewer 3 Report

In the paper 'Uniaxial Strain Dependence on Angle-resolved Optical Second Harmonic Generation from Indium Selenide Few-Layers' the authors presents their recent results about SHG from Indium Selenide layers. The topic is quite hot and the paper is interesting. The results are clear and well presents. I have only few minor issue which I list here:

1. I suggest to increase the legend of Figs.2,3.4,5 since they are very small;

2. I feel that a comparison with dielectric nanoantennas for SHG should be presented...

3. Some fundamental references about SHG in AlGaAs/GaAs monolithic nanoantennas should be cited in the paper...also because the SHG formalism come from that works.

Round 2

Reviewer 1 Report

The authors have substantially modified the manuscript and in the present variant it can be published. The only concern is to ask the authors to go through the text carefully and check some rough sayings.

Author Response

We appreciate Reviewer 1's careful review.  We have gone through the manuscript carefully and slightly revised some wordings. The "Track changes" function was used in the Word file.